# Evaluation of Pediatric Chronic Urticaria with Emphasis on Clinical and Laboratory Characteristics and Treatment Response to Omalizumab: A Real-Life Experience from a Tertiary Allergy Center

**DOI:** 10.3390/children11010086

**Published:** 2024-01-11

**Authors:** Aylin Kont Özhan, Tuğba Arıkoğlu

**Affiliations:** Department of Pediatric Allergy and Immunology, Faculty of Medicine, Mersin University, Mersin 33110, Turkey; arikoglutugba@mersin.edu.tr

**Keywords:** chronic spontaneous urticaria, children, chronic inducible urticaria, omalizumab, autoimmunity

## Abstract

Pediatric data on the clinical and etiologic features, treatment response, and use of omalizumab for chronic urticaria (CU) are quite limited. The aim of this study was to evaluate the clinical and demographic characteristics, laboratory findings, and response to treatment of CU in children. Children with a diagnosis of CU between 2019 and 2023 were included in the study. Information on demographic characteristics, clinical features, laboratory tests, provocation tests for inducible urticaria, urticaria activity scores (UAS7), and treatment responses were obtained from patients’ medical records. A total of 150 children (50.7% male) with CU were enrolled in the study. A total of 14 (9.3%) patients had autoimmune diseases of which 11 (7.3%) had autoimmune thyroiditis. Overall, 97 (64.7%) patients had chronic spontaneous urticaria (CSU) and 53 (35.3%) had chronic inducible urticaria. A total of 16 patients who remained symptomatic despite high-dose antihistamines were treated with omalizumab, with a good response in 13 (81.3%) and a partial response in 3 (18.7%) patients. CSU accounts for the majority of pediatric CU, with the etiology being in part related to an autoimmune background. This study provides an overview of CU in children and demonstrates the safety and efficacy of treatment with omalizumab.

## 1. Introduction

Chronic urticaria (CU) is a rare disease with a reported prevalence of 0.5–5% in the general population and 0.1–0.3% in childhood [1,2]. It can be frustrating for both patients and physicians as it is difficult to treat and can significantly impair the quality of life. CU is characterized by the appearance of hives and/or angioedema for a duration of six weeks or longer and is classified into two subgroups according to the underlying triggering factors of chronic spontaneous urticaria (CSU) and chronic inducible urticaria (CIU) [3]. Despite a link between CSU and autoimmune diseases, the exact etiology in CSU is unknown, whereas CIU is associated with various physical or chemical factors such as thermal triggers, ultraviolet radiation, mechanical factors, active and passive heating, and contact with certain substances [4].

Recent studies have shown that infections, parasites, food allergies, and autoimmunity are the possible causes of CU [5]. It has been suggested that infections may play a role in the etiology of CU. In recent years, an increasing number of studies have been published in the literature reporting an association between Helicobacter pylori and chronic urticaria in children [3]. In addition to that, it is difficult to determine the potential triggers of CSU, and little information is available on the triggers of CSU in childhood. CSU is considered to be mediated by functional autoantibodies to IgE itself or the high-affinity receptor for IgE. It is widely recognized that autoimmune mechanisms play a significant role in its immunopathogenesis [6]. Previous studies have shown that up to 38–45% of children with CU have a positive autologous serum skin test (ASST), which is used to identify autoimmune pathology [7,8]. According to the current classification, CIU entails a variety of physical and nonphysical urticarias. Physical urticarias include cold urticaria, dermographism, heat urticaria, delayed-pressure urticaria, solar urticaria, exercise-induced urticaria, and vibratory urticaria. On the other hand, nonphysical urticarias, such as contact urticaria, cholinergic urticaria, and aquagenic urticaria, are triggered by specific nonphysical stimuli [9,10].

CU has an unpredictable course and duration. Since the management of CSU can be challenging and burdensome for patients due to its chronic nature and impact on daily life, individualized treatment is crucial [11]. The current guideline on CU recommends a stepwise therapeutic approach whereby second-generation H1 antihistamines (sgAHs) given at the standard dose are employed as the first-line treatment and the daily dose can be increased up to fourfold the standard dosage to provide disease control. Omalizumab, a recombinant, humanized, monoclonal anti-IgE antibody, is recommended for add-on therapy in CU when sgAHs fail to achieve the appropriate response. In addition, cyclosporine can be effective in patients with severe CU who remain inadequately controlled using other therapy modalities [3,12,13].

Data on the clinical and etiologic characteristics, persistence, treatment, and prognosis of CU in children are limited. Therefore, the aim of the present study was to evaluate the clinical and demographic characteristics, laboratory results, treatment response, and natural course of CU in children.

## 2. Materials and Methods

Patients who were diagnosed with CU and subsequently followed up on in the Pediatric Allergy and Immunology Department of Mersin University between 2019 and 2023 were included in the study. CU was defined as the presence of urticaria and/or angioedema lasting for more than six weeks [3]. Medical charts of the children were retrospectively reviewed to determine clinical and demographic characteristics, including age, gender, symptom onset, duration of symptoms, triggering factors, personal atopy, accompanying allergic and chronic diseases, presence of angioedema, a family history of atopy, allergic diseases, chronic urticarias, or autoimmune diseases, and response to therapeutic agents. The definition of controlled with treatment refers to patients whose urticaria symptoms were effectively managed and kept under control through the use of therapy. Remission was defined as the absence of symptoms for at least six months, without the need for ongoing treatment [14]. The urticaria activity score 7 (UAS7) was used to assess disease activity. It ranges from 0 to 42, with higher scores suggesting greater disease activity. Response to treatment was classified as follows: complete response (UAS7 score = 0), good response (1 ≤ UAS7 score < 7), partial response (7 ≤ UAS7 score ≤ 15), and poor response (UAS7 score ≥ 16) [15].

Laboratory tests including complete blood cell count, absolute eosinophil count, absolute basophil count, liver enzymes, kidney function tests, erythrocyte sedimentation rate (ESR), C-reactive proteins (CRP), thyroid function tests, antithyroglobulin and antithyroid peroxidase autoantibodies, antinuclear antibodies (ANAs), hepatitis serology, levels of complement C3 and C4, a stool examination for parasites and Helicobacter pylori antigen, urine analysis, and throat and urine cultures were evaluated. Eosinopenia was defined as <0.05 × 10^9^/L and basopenia as <0.01 × 10^9^/L [16]. Results of atopy tests, ASSTs and diagnostic provocation tests for physical urticaria were recorded. The study was approved by the Institutional Ethics Committee of Mersin University (study number: 2023/511). Patient consent was obtained prior to the start of the study.

### 2.1. Atopy Tests

Atopic sensitization was determined by serum total IgE, inhalant and food spesific IgE, and skin prick tests. All patients underwent skin prick tests with the following inhalant and food antigens: Dermatophagoides pteronyssinus, Dermatophagoides farina, Alternaria Alternata, mixed grass, ragweed, tree pollen, animal allergens including cat and dog dander, cockroach, cow’s milk, egg, peanut, tree nut, soy, and wheat in addition to histamine and saline controls. A mean diameter of the wheal that was at least 3 mm larger than the negative control and accompanied by surrounding erythema was indicative of a positive skin prick test. Atopy was defined as specific IgE (>0.35 kU/L) or skin prick test positivity to any food or inhalant allergen. In the case of a suspected food allergy, an oral food provocation test was performed after an elimination diet to confirm the diagnosis.

### 2.2. Autologous Serum Skin Test

ASST is a valuable diagnostic procedure in the evaluation of CU. During the test, 0.05 mL of sterile, fresh autologous serum was applied through intradermal injection on the volar surface of the forearm. Additionally, histamine (10 µg/mL) and 0.9% saline solution were used as positive and negative controls, respectively. Wheal and flare response was evaluated after 30 min. A positive reaction was defined as a mean diameter of the induration at the serum injection site at least 1.5 mm greater when compared with the negative control [17].

### 2.3. Diagnostic Physical Urticaria Provocation Tests

Based on the EAACI guidelines, specific provocation tests were performed to diagnose CIU [3,18]. The presence of dermographism was evaluated by stroking the skin, typically on the volar forearm, with a tongue depressor. If a wheal and flare reaction occurred within 10 min after stroking the skin then the test was considered positive. For cholinergic urticaria, an activity such as a bicycle ergometer exercise was performed to induce sweating. The typical appearance of urticaria within minutes after exercise indicated cholinergic urticaria. The presence of delayed pressure urticaria was assessed by placing a 5 kg rod across the patient’s forearms for 20 min. The patient was observed for skin symptoms over the next 24 h. The emergence of an erythematous palpable swelling that developed 4 to 6 h after applying pressure was considered a positive result. The diagnosis of aquagenic urticaria was made if the application of wet compression at the body temperature of water (35 °C) to the upper body for 30 min resulted in urticarial reaction. Cold urticaria was diagnosed through the provocation test in which an ice cube in a thin plastic bag was placed on the volar surface of the forearm for 5–10 min. A positive test was indicated through the appearance of a localized wheal and flare reaction upon rewarming. The presence of solar urticaria was determined by observing urticaria episodes on the skin after exposure to sunlight or specific wavelengths of ultraviolet A and ultraviolet B [19,20].

### 2.4. Statistical Analysis

Statistical analysis was performed using Jamovi version 2.3.26. Categorical endpoints were summarized as percentages. Because continuous variables were not normally distributed according to the result of the Kolmogorov–Smirnov test, descriptive analyses were performed using median values (25–75 percentiles). The Wilcoxon test was used to compare the UAS7 scores at baseline and after six months of omalizumab treatment. A *p*-value of less than 0.05 was considered statistically significant.

## 3. Results

A total of 150 children with CU were included in the study, with a median age of 12.0 years (IQR: 7.0–16.0 years) at admission and a predominance of males (50.7%). The median age upon the onset of symptoms was 10.0 years (IQR: 6.0–14.0 years), and the median duration of symptoms before treatment was 15.0 months (IQR: 9.0–28.5 months). Overall, 23 (15.3%) children complained of CU accompanied by angioedema. Forty-four patients had a history of allergic diseases, most commonly asthma, and 24 (16.0%) children had other chronic diseases (autoimmune diseases, familial Mediterranean fever, reflux and/or gastritis, growth hormone deficiency, recurrent urine infections) accompanied by CU, including 14 (9.3%) with autoimmune diseases (autoimmune thyroiditis, juvenile rheumatoid arthritis, diabetes mellitus, acute rheumatic fever). A family history of allergic diseases, chronic urticaria, and autoimmunity was present in 35 (23.3%), 11 (4.4%), and 16 (10.7%) patients, respectively. Of all patients with CU, 97 (64.7%) had CSU and 53 (35.3%) had CIU. The clinical and demographic characteristics of the patients were given in Table 1.

Regarding laboratory parameters, 16 out of 149 (10.7%) patients had eosinopenia and 38 out of 147 (25.9%) patients had basopenia. Of the 132 patients with CU, 77 (58.3%) had elevated IgE levels. Food- and inhalant-specific IgE forms of positivity were detected in 20 (13.3%) and 29 (19.3%) of all patients, respectively. A positive skin prick test for inhalant allergens, mostly due to dust mites, was present in 25 out of 134 (18.6%) patients. A food allergy was confirmed using an oral food provocation test only in one patient with a nut allergy. Twenty-four out of 124 (19.4%) children had ANA positivity, but none of the patients were diagnosed with rheumatologic disease during the follow-up period. Antithyroid antibodies, antithyroid peroxidase, and antithyroglobulin were positive results were found in 13 out of 134 (9.7%) patients and 11 patients had abnormal thyroid function tests. While a complete response of CU was obtained in one out of nine patients treated for autoimmune thyroiditis, three patients had a partial response and the remaining five patients had no response. A positive ASST result was found in 18 out of 83 (21.7%) children. Out of 131 patients, 12 (9.2%) had a positive stool examination for Helicobacter pylori. All of these patients were treated for Helicobacter pylori, and only two had a complete remission of CU, while 3 patients had a partial response and the remaining 7 children had no response. A total of 5 (3.5%) out of 144 patients had a positive stool examination for parasites, 3 with Giardia intestinalis and 2 with Enterobius vermicularis. Antiparasitic therapy was given to these five cases of which two showed a partial response after treatment, whereas the others did not benefit. Abnormal urine results (leukocyturia, erythrocyturia) were obtained in 18 (12%) out of 144 patients. Urine cultures could be obtained in 13 of the patients with abnormal urine analysis (3 of these patients had symptoms such as dysuria, frequent urination, and abdominal pain). Two of the three clinically symptomatic patients had positive urine culture results (one with Klebsiella and the other with Escherichia Coli) and appropriate antibiotic treatment was given. The remaining 16 patients with abnormal urine analysis had transient pyuria or erythrocyturia with no clinical significance. No patient had kidney disease. The results of the laboratory findings are shown in Table 2.

Out of the 53 (35.3%) patients with CIU based on diagnostic provocation tests, the most common type was symptomatic dermographism (27.3%), which was followed by cholinergic urticaria (17.6%) and delayed pressure urticaria (7.8%). Three patients were diagnosed with cold urticaria, one with solar urticaria, and one with aquagenic urticaria (Table 2). A significant predominance of males was present in the CSU group compared with patients with CIU (*p* = 0.019). However, there was no significant difference between CSU and CIU in terms of age at admission, presence of atopy, and family history of CU as well as autoimmunity, serum-total IgE levels, presence of eosinopenia and basopenia, ANA positivity, existence of thyroid antibodies, positive autologous serum skin test, and Helicobacter pylori positivity. The comparison of the clinical and laboratory findings of patients according to chronic urticaria subtypes are shown in Table 3.

Out of all patients, 111 achieved clinical remission and 39 remained under control through the appropriate CU treatment. A total of 100 patients (66.6%) with CU were successfully treated with standard doses of sgAHs (86 in clinical remission and 14 under control with treatment). Thirty-four patients (22.7%) required higher doses of sgAHs and sixteen patients (10.7%) needed omalizumab in combination with higher dose sgAHs. Only one patient with a partial response to omalizumab in combination with higher dose antihistamines was successfully treated with cyclosporine (3 mg/kg/day).

A total of 16 patients were treated with omalizumab at a starting dose of 300 mg every four weeks, with a complete or good response in 13 (81.3%) patients and partial response in 3 (18.7%). The median time to achieve a complete or good response was 6.0 (IQR 4.5–7.0) weeks among 13 children. The median UAS7 scores (IQR) at baseline and after six months of therapy were available for all 16 children, and the median UAS7 score at baseline decreased significantly from 32.0 (29.0–35.8) to 3.0 (0.3–4.8) after omalizumab treatment (*p* < 0.01). Omalizumab was continued in 9 patients, with a good response in 6 and a partial response in 3 patients. In two out of these three patients with a partial response, urticaria was controlled by increasing the omalizumab dose to 300 mg every two weeks, while the remaining patient had a partial response despite high-dose omalizumab treatment but achieved a good response by adding cyclosporine to the high-dose omalizumab. At the point of the follow-up, a total of seven patients were able to discontinue omalizumab treatment, three of whom experienced a relapse, while the remaining four maintained remission after treatment discontinuation. The three patients with a relapse, who are described below, had a complete response after the reinitiation of omalizumab treatment. Patient 5 (Table 4) received omalizumab treatment for 31 months and his complaints started again 11 weeks after the treatment was stopped. Patient 10 (Table 4) received omalizumab treatment for 27 months and his complaints started 13 weeks after the treatment was stopped. Patient 15 (Table 4) was given omalizumab treatment for 23 months and his complaints started again 14 weeks after the treatment was stopped. All these three patients were retreated using omalizumab at a dose of 300 mg/4 weeks and had been receiving this treatment for four months. No adverse effects were observed during treatment with omalizumab or cyclosporine (Table 4, Figure 1).

## 4. Discussion

CU is indeed a heterogeneous disease that is characterized by a wide range of clinical characteristics, demographic features, and natural courses [21]. Unfortunately, data regarding phenotypes/endotypes, treatment response, and prognosis of CU in the pediatric population is quite limited. In the present study, we evaluated clinical and etiologic features, laboratory findings, treatment response, and natural courses among 150 children with CU.

Although there have been several studies demonstrating an association between atopy, allergic diseases, and CU, the exact nature of the relationship between CU and atopy is not completely clear [14,22]. The reported prevalence of atopy was 16.7–33.7% in children with CU [23,24,25]. In the present study, 29.3% of patients had a personal history of allergic disease, most commonly asthma, and 23% of patients had a family history of atopy/allergic disease. A positive skin prick test for inhalant allergens, mostly for dust mites, was present in 18.6% of patients. Food- and inhalant- specific forms of IgE positivity were detected in 13.3% and 19.3% of all patients, respectively. However, a food allergy was confirmed through an oral food provocation test in only one patient with a nut allergy. A previous study by Park et al. reported that children with CSU had a higher prevalence of food and inhalant sensitization. Of note, the authors did not perform oral provocation tests to confirm food allergies but only assessed sensitization to foods [22]. Previous studies demonstrated that CD63 expression and spontaneous basophil activation were significantly higher in CU patients with allergic sensitization than in children without it, suggesting that allergic sensitization may be an underlying factor contributing to the pathogenesis of CU [26,27]. Another previous study reported a higher prevalence of dust mite sensitization and multiple sensitizations in children with CU but no significant difference in the prevalence of food sensitization [28]. These studies also highlighted that food sensitization was associated with inadequate disease control and treatment response to CU.

Despite limited data focusing on the pediatric population, a growing number of studies have investigated the role of autoimmunity in the pathogenesis of CU [29,30]. The emergence of circulating autoantibodies to IgE or the high-affinity IgE receptor leading to mast cell degranulation are underlined as the most highlighted mechanism in pathogenesis. In addition, patients with CU are known to be at an increased risk of developing other autoimmune diseases, so clinicians should be aware of this coexistence and monitor patients periodically for the development of autoantibodies [31,32]. In the present study, 14 patients (9.3%) had autoimmune diseases, including autoimmune thyroiditis (n = 11), diabetes mellitus (n = 1), and juvenile rheumatoid arthritis (n = 1). A complete response of CU was obtained in one out of nine patients treated for autoimmune thyroiditis, whereas three patients showed a partial response and the remaining five patients did not respond. These things considered, intradermal injection of autologous serum can elicit a wheal-and-flare response in some patients with CSU, thus confirming a positive ASST test. The positive ASST test indicates the presence of functional circulating autoantibodies. The prevalence of ASST positivity in children with CU has been reported to range from 38.3% to 53.5% [8,23,33]. In our cohort, a positive ASST result was found in 21.7% of children, suggesting an autoimmune etiology in CU. Previous studies conducted in children with CU reported the presence of ANA positivity in the range of 13.8 to 23% [8,23]. In the present study, 19.4% of children had ANA positivity. These patients were evaluated for rheumatologic disease, but none of them were diagnosed with rheumatologic or connective tissue disease during the follow-up period.

There is limited and conflicting evidence regarding the role of Helicobacter pylori in CU. The reported rate of Helicobacter pylori positivity in children with CU ranges from 9.7% to 32.8% [23,33]. In the present study, 9.2% of patients had a positive stool examination for H. pylori and, after eradication treatment for H. pylori, only two had a complete remission of CU, while three patients had a partial response and the remaining seven children had no response. Parasitic infections are considered another potential cause of CU, although there is no clear, consistent link between both entities. Only a small series of cases has demonstrated an association between CU and parasites such as Blastocystis hominis, Giardia lamblia, Strongyloides stercoralis, and Toxocara canis [34,35]. In a previous study, 5.3% of children with CU had parasites in their stool, but antiparasitic therapy did not contribute to CU remission. In this regard, five of the patients in our study had positive stool examination for parasites, three with Giardia intestinalis and two with Enterobius vermicularis. Antiparasitic therapy was given in these five cases; two showed a partial response after treatment, whereas the others did not benefit.

CU is classified as CSU and CIU based on the underlying causes or eliciting factors [3,4]. CSU is the more common type of CU, as reported in the literature [33,36]. In a study of 117 children diagnosed with CIU, symptomatic dermographism (65%) was the most common type of CIU, which was followed by cold urticaria (17%) and cholinergic urticaria (15.4) [37]. Overall, 35.3% of patients in the present study were diagnosed with CIU based on diagnostic provocation tests, with symptomatic dermographism (27.3%) being the most common type, which was followed by cholinergic urticaria (17.6%) and delayed pressure urticaria (7.8%). Three patients were diagnosed with cold urticaria, one with solar urticaria, and one with aquagenic urticaria. A significant predominance of males was present in the CSU group compared with patients with CIU. However, there was no significant difference between CSU and CIU in terms of clinical characteristics and laboratory findings.

The most recent EAACI guideline recommends sgAHs at standard doses as a first-line choice and the up-dosing of sgAHs as second-line therapy for refractory patients with CU. Omalizumab has been recommended as a third-line therapy for the treatment of CU in adults and adolescents aged 12 years and older who do not respond adequately to sgAHs [38]. After follow-up in our department, a total of 100 patients (66.6%) with CU were successfully treated with standard doses of sgAHs, whereas 34 patients (22.7%) required higher doses of sgAHs and 16 patients (10.7%) required omalizumab in combination with higher doses sgAHs. Only one patient treated with omalizumab in combination with higher doses of sgAHs achieved a complete response with the addition of cyclosporine.

Omalizumab is a recombinant monoclonal antibody that specifically targets and binds to human IgE, thereby preventing free IgE from binding to its high-affinity receptor (FcƐRI) on mast cells and basophils. This mechanism ultimately leads to a decrease in serum IgE levels, FcƐRI receptors on basophils, and the degranulation of mast cells/basophils [3,15]. Although experience on the safety and efficacy of omalizumab in children with CU is limited and based mainly on a few clinical case series [39,40,41], it has been shown to be safe and efficient in large randomized controlled trials in children with asthma [42,43]. In the present study, a total of 16 patients were treated with omalizumab, with a complete or good response in 13 (81.3%) and a partial response in 3 (18.7%) patients. In two of the three patients with a partial response, urticaria was controlled by increasing the omalizumab dose by 300 mg every two weeks. The median UAS7 scores at baseline and after six months of therapy were available for children treated with omalizumab and the median UAS7 score at baseline decreased significantly from 32.0 to 3.0 after treatment. Consistent with emerging data, the results of our study suggest that omalizumab appears to be highly efficient and promising in the treatment of children with CU. Prolonged treatment intervals or tapered dosage before discontinuation were tried in seven patients, and four patients achieved remission after treatment discontinuation. However, relapses occurred in three patients and the reintroduction of omalizumab treatment achieved a complete response. No adverse effects were observed during treatment with omalizumab, suggesting that it can be used as a safe therapeutic agent for children with CU. Thus, the safety outcomes of the current study are consistent with the results of other studies on children with CU who were treated with omalizumab [40,41]. In patients who have a partial response to omalizumab and in whom complete control cannot be achieved with the up-dosing of omalizumab, combining cyclosporine treatment with omalizumab should be considered [3,6]. In this regard, one of our patients had a partial response despite high-dose omalizumab treatment but achieved a good response with the additional administration of cyclosporine.

In recent years, much attention has been directed towards reliable biomarkers in an attempt to predict disease severity, prognosis, and also response to treatment strategies [14,44,45]. Among these parameters, female gender, comorbid autoimmune disease, higher UAS7 scores at baseline, decreased eosinophil and basophil counts, lower total IgE levels (<40 IU/mL), a positive ASST test, or a positive basophil activation test seem to be the most interesting markers [16,44,46]. A relationship between the response to omalizumab and the aforementioned parameters could not be established in the present study due to the small number of patients treated with omalizumab, although all patients with a relapse or a partial response showed higher baseline UAS7 scores. In addition, all patients with a relapse had eosinopenia and/or basopenia. Lower total IgE levels (<40 IU/mL) were detected in one of the patients with a relapse and two of the patients with a partial response, supporting the notion that better response rates can be expected in patients with higher baseline total IgE levels. Moreover, one patient with a relapse had autoimmune thyroiditis and the other patient with a partial response had positive ASST and ANA test results in addition to a diagnosis of autoimmune thyroiditis.

One of the limitations of our study was its retrospective nature, which may have led to some missing data in the medical charts. Another limitation was the relatively small number of patients treated with omalizumab. In addition, follow-up results could not be obtained for all patients. However, we used the UAS7 score as an objective marker to determine the disease severity and response to omalizumab treatment. Despite these limitations, our study represents real-life data from the practice of a tertiary allergy center on clinical and etiologic features and response rates to treatment in CU, thereby providing valuable insight into the natural course of CU in children.

## 5. Conclusions

Chronic spontaneous urticaria constitutes the majority of pediatric CUs and the etiology is partly related to an autoimmune basis. In addition, clinicians should be aware of CIU and, when it is suspected, perform diagnostic provocation tests to accurately identify specific triggers and treat these patients accordingly. This study also supported the safety and efficacy of omalizumab treatment in children with CU. Future large studies are needed to elucidate the clinical and laboratory features and underlying immunologic mechanisms of CU, with a focus on potential reliable biomarkers in order to better understand endotypes/phenotypes in children and predict disease severity, treatment response, and prognosis.

## Figures and Tables

**Figure 1 children-11-00086-f001:**
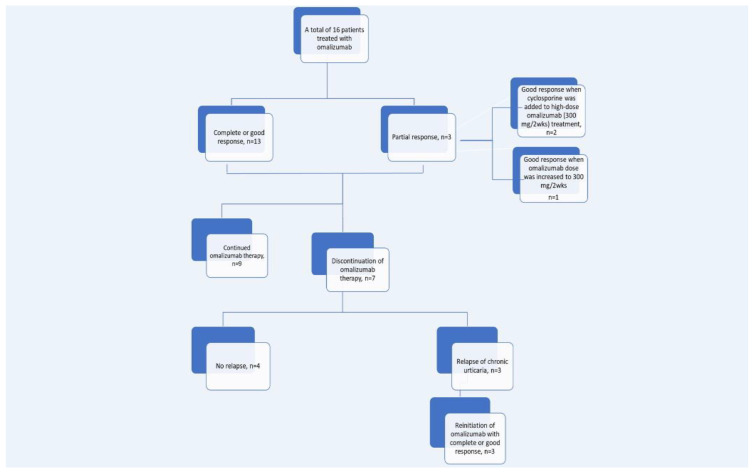
Schematic data of patients treated with omalizumab.

**Table 1 children-11-00086-t001:** Clinical and demographic characteristics of patients with chronic urticaria.

Variable	Patients with Chronic Urticaria (n = 150) %
Age at admission (years), median (IQR)	12.0 (7.0–16.0)
Age at onset of symptoms, (years) median (IQR)	10.0 (6.0–14.0)
Duration of symptoms, (months) (IQR)	15.0 (9.0–28.5)
Male gender	76.0 (50.7)
Presence of angioedema	22.0 (14.7)
Personal history of allergic diseases	44.0 (29.3)
Atopic dermatitis	1.0 (0.7)
Allergic rhinitis	6.0 (4.0)
Asthma	26.0 (17.3)
Asthma with allergic rhinitis	8.0 (5.3)
Food allergy	1.0 (0.7)
NSAID allergy	2.0 (1.3)
Concomitant diseases	24.0 (16.0)
Autoimmune thyroiditis	11.0 (7.3)
Juvenile rheumatoid arthritis	1.0 (0.7)
Diabetes mellitus	1.0 (0.7)
Acute rheumatic fever	1.0 (0.7)
Famialial Mediterranean Fever	1.0 (0.7)
Reflux and/or gastritis	5.0 (3.3)
Growth hormone deficiency	1.0 (0.7)
Recurrent urine infections	3.0 (2.0)
Family history of allergy and autoimmunity	
Atopy/allergic diseases	35.0 (23.3)
Chronic urticaria	11.0 (4.4)
Autoimmunity	16.0 (10.7)
Chronic spontaneous urticaria	97.0 (64.7)
Chronic inducible urticaria	53.0 (35.3)

IQR: interquartile range; NSAID: nonsteroidal anti-inflammatory drug.

**Table 2 children-11-00086-t002:** Laboratory findings for patients with chronic urticaria.

Variable	Patients n (%)
Peripheral eosinophil count (×10^3^/µL), median (IQR)	170.0 (90.0–300.0)
Eosinophilia > 4%	22/149 (14.8)
Presence of eosinopenia	16/149 (10.7)
Peripheral basophil count (×10^3^/µL), median (IQR)	30.0 (10.0–50.0)
Presence of basopenia	38/147 (25.9)
Serum-total IgE levels (IU/mL), median (IQR)	116.5 (51.3–201.7)
Increased total IgE levels, median (IQR)	77/132 (58.3)
Food-specific IgE positivity (>0.35 kU/L)	20/150 (13.3)
Inhalant-specific IgE positivity (>0.35 kU/L)	29/150 (19.3)
Skin prick test positivity for food allergen	9/98 (9.2)
Skin prick test positivity for inhalant allergen	25/134 (18.6)
House dust mite	18/134 (13.4)
Pollen	13/134 (9.7)
Mold	3/134 (2.2)
Animal dander	2/134 (1.5)
Cockroach	2/134 (1.5)
Increased CRP value (0.5 g/dL)	21/143 (14.7)
Increased ESR value (>20 mm/hr)	8/143 (5.6)
Increased transaminase level	6/149 (4.0)
Abnormal urine analysis	18/144 (12.5)
Positive stool examination for parasites	5/144 (3.5)
Antinuclear antibody (ANA) positivity	24/124 (19.4)
Abnormal complement, C3 or C4	7/98 (7.1)
Positive hepatitis serology	0/92 (0.0)
Abnormal thyroid function test	11/146 (7.5)
Positive thyroid antibodies	13/134 (9.7)
Helicobacter pylori positivity	12/131 (9.2)
Positive autologous serum skin test	18/83 (21.7)
Positive throat culture	0/70 (0.0)
Positive urine culture	2/13 (15.4)
Results of provocation tests in chronic inducible urticaria	53.0 (35.3)
Dermographism	41/150 (27.3)
Cholinergic urticaria	15/85 (17.6)
Delayed pressure urticaria	4/51 (7.8)
Cold urticaria	3/74 (4.1)
Solar urticaria	1/50 (2.0)
Aquagenic urticaria	1/72 (1.4)

IQR: interquartile range; CRP: C-reactive protei; ESR: erythrocyte sedimentation rate.

**Table 3 children-11-00086-t003:** Comparison of clinical and laboratory findings of patients according to chronic urticaria subtypes.

Variable	CSU (n: 97) n (%)	CIU (n: 53) n (%)	*p* Value
Age at admission (years), median (IQR)	11.0 (7.0–16.0)	12.0 (8.5–15.5)	0.426
Male gender	56 (57.7)	20 (37.7)	0.019
Presence of angioedema	16 (16.5)	7 (13.2)	0.593
Personal atopy	17 (17.5)	14 (26.4)	0.199
Family history of chronic urticaria	9 (9.3)	2 (3.8)	0.329
Family history of autoimmunity	13 (13.4)	3 (5.7)	0.142
Eosinophils > 4%	17 (17.7)	5 (9.4)	0.173
Presence of eosinopenia	12 (12.5)	4 (7.5)	0.350
Presence of basopenia	25 (26.0)	13 (25.5)	0.942
Serum-total IgE levels (IU/mL), median (IQR)	117.0 (51.0–175.0)	114.0 (55.5–355.0)	0.455
Positive thyroid antibodies	7 (8.0)	6 (13.0)	0.368
Positive autologous serum skin test	11 (21.2)	7 (20.0)	0.896
Helicobacter pylori positivity	8 (8.5)	4 (10.8)	0.739
Antinuclear antibody (ANA) positivity	13 (16.0)	11 (25.6)	0.201
Response to antihistaminic treatment Remisssion Controlled with treatment	71 (73.2) 26 (26.8)	40 (75.5) 13 (24.5)	0.761

IQR: interquartile range.

**Table 4 children-11-00086-t004:** Data of patients treated with omalizumab.

Patient No	Sex/Age (ys)	CU Type	Allergic Comorbidities	IgE Level IU/mL	UAS7 Score at Baseline	UAS7 Score at Six Months of Therapy	Omalizumab Dose	Withdrawal of Therapy	Response to Treatment	Relapse	Current Dose
1	M/14	CSU	Asthma, allergic rhinitis	257	29	3	300 mg/4 wks/7 mo	No	Good	No	300 mg/4 wks
2	M/12	CSU	None	21	42	3	300 mg/4 wks/8 mo	No	Good	No	300 mg/4 wks
3	M/15	CSU	Food allergy	50	34	10	300 mg/4 wks/6 mo 300 mg/2 wks/2 mo	No	Partial	No	300 mg/2 wks
4	F/17	CSU	Asthma, allergic rhinitis	169	29	3	300 mg/4 wks/14 mo 150 mg/4 wks/10 mo	No	Good	No	150 mg/4 wks
5	M/17	CSU	None	126	31	0	300 mg/4 wks/15 mo 300 mg/6 wks/6 mo 300 mg/8 wks/10 mo	Yes	Complete	Yes	300 mg/4 wks
6	F/18	CSU	None	174	30	2	300 mg/4 wks/11 mo 300 mg/6 wks/6 mo 300 mg/8 wks/6 mo	Yes	Good	No	-
7	F/18	CSU	None	150	35	5	300 mg/4 wks/9 mo 300 mg/8 wks/3 mo 150 mg/8 wks/3 mo	No	Good	No	150 mg/8 wks
8	M/17	CSU	Asthma	28	42	15	300 mg/4 wks/6 mo 450 mg/4 wks/4 mo 600 mg/4 wks/13 mo 300 mg/2 wks/8 mo	No	Partial response to high-dose omalizumab	No	300 mg/2 wks and cyclosporine
9	F/18	CSU	None	52	27	0	300 mg/4 wks/4 mo 300 mg/6 wks/9 mo 300 mg/8 wks/6 mo	Yes	Complete	No	-
10	F/16	CSU	None	110	30	3	300 mg/4 wks/8 mo 300 mg/6 wks/9 mo 300 mg/8 wks/10 mo	Yes	Good	Yes	300 mg/4 wks
11	M/17	CSU	Asthma	80	28	3	300 mg/4 wks/6 mo 150 mg/4 wks/4 mo 150 mg/8 wks/6 mo	No	Good	No	150 mg/8 wks
12	F/18	CSU	None	318	25	1	300 mg/4 wks/6 mo 300 mg/6 wks/6 mo 300 mg/8 wks/8 mo	Yes	Good	No	-
13	F/12	CIU	None	5	35	12	300 mg/4 wks/4 mo 450 mg/4 wks/6 mo 300 mg/2 wks/6 mo	No	Partial	No	300 mg/2 wks
14	F/18	CIU	Asthma	202	33	0	300 mg/4 wks/9 mo 300 mg/6 wks/6 mo 300 mg/8 wks/8 mo	Yes	Complete	No	-
15	F/14	CIU	None	17	36	0	300 mg/4 wks/8 mo 300 mg/6 wks/9 mo 300 mg/8 wks/6 mo	Yes	Complete	Yes	300 mg/4 wks
16	F/16	CSU	Asthma	315	38	5	300 mg/4 wks/8 mo	No	Good	No	300 mg/4 wks

CU: chronic urticaria; CIU: chronic inducible urticaria; CSU: chronic spontaneous urticaria; F: female; M: male; UAS7: urticaria activity score, wks: weeks, month: mo.

## Data Availability

The data presented in this study are available on request from the corresponding author. The data are not publicly available due to privacy and the ethical restrictions.

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
