# Peer review of "Evaluation of Pediatric Chronic Urticaria with Emphasis on Clinical and Laboratory Characteristics and Treatment Response to Omalizumab: A Real-Life Experience from a Tertiary Allergy Center"

_children, 2024, doi:10.3390/children11010086_

Round 1

Reviewer 1 Report

Comments and Suggestions for Authors

Review of the manuscript: children-2793669

The manuscript is a valuable contribution to the knowledge on pediatric aspects of CU. It provides detailed, thoroughly described data on 150 children assessed for urticaria in a tertiary center in Türkiye. It is worth noting that the authors have undertaken the effort to pursue a meticulous diagnostic process in order to identify possible types of urticaria, in particular with regard to CIU. The diagnostic and provocation tests are correctly applied and described and they explain the basis on which patients have been attributed to a given diagnostic group. Also, the accounts on the parasitic infestations concomitant with urticaria are of high interest. Incomplete resolution of urticaria after anti-parasitic treatment in the substantial percentage of cases is the experienced shared by many centers and individual practices and it was interesting to find its confirmation also in a geographically distant area.

In what regards other analyses: it is reported that 18 patients had abnormal urine analysis and 2  had positive urine culture. Would it be possible to retrieve data and give a brief account what were the abnormalities in the urine analysis? Have they been associated with kidney pathology or other abnormalities? Regarding positive urine cultures: as it can be presumed form the table, urine culture was performed only in 13 patients, of whom 2 had positive result. What was the criterion for referring the patient for the urine culture? Please provide – if available – further information if these patients were symptomatic and what pathogens were identified in 2 patients with positive culture. Finally, what about the remaining 11 – if they were symptomatic, was the reason for their symptoms ascertained whatsoever?

The results are correctly discussed and findings are reflected in the DISCUSSION section. The authors are aware of the limitations of their study and address them critically in their summary. Apart from the remarks presented above I don’t have any other comments regarding this manuscript.

I recommend proceeding with its publication process after the issues raised in my review are addressed and implemented or rebutted.

Author Response

REVIEWER 1:

QUESTION 1: In what regards other analyses: it is reported that 18 patients had abnormal urine analysis and 2 had positive urine culture. Would it be possible to retrieve data and give a brief account what were the abnormalities in the urine analysis? Have they been associated with kidney pathology or other abnormalities? Regarding positive urine cultures: as it can be presumed form the table, urine culture was performed only in 13 patients, of whom 2 had positive result. What was the criterion for referring the patient for the urine culture? Please provide – if available – further information if these patients were symptomatic and what pathogens were identified in 2 patients with positive culture. Finally, what about the remaining 11 – if they were symptomatic, was the reason for their symptoms ascertained whatsoever?

ANSWER 1: Thank you for the suggestion. Urine analysis was also evaluated among routine diagnostic tests. Abnormal urine results (leukocyturia, erythrocyturia) were obtained in 18 (12%) of 144 patients. Urine culture could be obtained in 13 of the patients with abnormal urine analysis (3 patients had symptoms such as dysuria, frequent urination, and abdominal pain). Two of the three clinically symptomatic patients had positive urine culture results (one with Klebsiella and the other with Escherichia Coli) and appropriate antibiotic treatment was given. The remaining 16 patients with abnormal urine analysis had transient pyuria or erythrocyturia with no clinical significance. No patient had kidney disease.

Reviewer 2 Report

Comments and Suggestions for Authors

Manuscript ID: children-2793669
Type of manuscript: Article
Title: Evaluation of pediatric chronic urticaria with emphasis on clinical
and laboratory characteristics and treatment response to omalizumab: A real-life experience from a tertiary allergy center

The abstract summarizes the article and the purpose of the study well.

The introduction adequately illustrates the structure of the article.

The “materials and methods” used are very detailed and clearly described.

Line 87 Please explain the abbreviation ASST here, as it is the first time when it is found in the text

In the “Results” section I found extremely interesting the approach of unresponsive patients to classical therapies and the evolution of urticaria on omalizumab.

Line 200: after how many weeks, the complete or good response was achieved in those 13 children?

Good idea to illustrate the results in Figure 1, but I am not able to see the whole picture (maybe there is something with my soft?). However, as much as I see, it looks ok.

Line 211 How long have you continued the treatment with omalizumab for the 3 patients with relapsed urticaria and reinitiation of omalizumab treatment?

Overall, the paper is interesting and provides an important advance in a very specific topic, pediatric chronic urticaria. The messages delivered are clear and of great importance to the clinicians.

Author Response

REVIEWER 2:

QUESTION 1: Line 87 Please explain the abbreviation ASST here, as it is the first time when it is found in the text

ANSWER 1: Thank you for the suggestion. Since this word was previously written in long form in line 35, it is stated as an abbreviation (ASST).

QUESTION 2: Line 200: after how many weeks, the complete or good response was achieved in those 13 children?

ANSWER 2: Thank you for the suggestion. The median time to achieve a complete or good response was 6.0 (IQR 4.5-7.0) weeks in 13 children.

QUESTION 3: Good idea to illustrate the results in Figure 1, but I am not able to see the whole picture (maybe there is something with my soft?). However, as much as I see, it looks ok.

ANSWER 3: Thank you for the suggestion. Figure 1 was revised according to the journal's writing rules.

QUESTION 4: Line 211 How long have you continued the treatment with omalizumab for the 3 patients with relapsed urticaria and reinitiation of omalizumab treatment?

ANSWER 4: Thank you for the suggestion.

Patient 5 (Table 4); The patient received omalizumab treatment for 31 months, and his complaints started again 11 weeks after the treatment was stopped. Patient 10 (Table 4); Omalizumab treatment was administered for 27 months, and his complaints started 13 weeks after the treatment was stopped. Patient 15 (Table 4); The patient was given omalizumab treatment for 23 months, and his complaints started again 14 weeks after the treatment was stopped. All these 3 patients were retreated with omalizumab with a dose of 300 mg/4 wks and has been receiving this treatment for 4 months.
